# Splicing Enhancers at Intron–Exon Borders Participate in Acceptor Splice Sites Recognition

**DOI:** 10.3390/ijms21186553

**Published:** 2020-09-08

**Authors:** Tatiana Kováčová, Přemysl Souček, Pavla Hujová, Tomáš Freiberger, Lucie Grodecká

**Affiliations:** 1Molecular Genetics Laboratory, Centre for Cardiovascular Surgery and Transplantation, 656 91 Brno, Czech Republic; tatkov@cktch.cz (T.K.); presou@cktch.cz (P.S.); pavhuj@cktch.cz (P.H.); tomas.freiberger@cktch.cz (T.F.); 2Faculty of Medicine, Masaryk University, 625 00 Brno, Czech Republic

**Keywords:** pre-mRNA splicing, splicing enhancer, U2AF35, acceptor splice site recognition, SRSF1

## Abstract

Acceptor splice site recognition (3′ splice site: 3′ss) is a fundamental step in precursor messenger RNA (pre-mRNA) splicing. Generally, the U2 small nuclear ribonucleoprotein (snRNP) auxiliary factor (U2AF) heterodimer recognizes the 3′ss, of which U2AF35 has a dual function: (i) It binds to the intron–exon border of some 3′ss and (ii) mediates enhancer-binding splicing activators’ interactions with the spliceosome. Alternative mechanisms for 3′ss recognition have been suggested, yet they are still not thoroughly understood. Here, we analyzed 3′ss recognition where the intron–exon border is bound by a ubiquitous splicing regulator SRSF1. Using the minigene analysis of two model exons and their mutants, *BRCA2* exon 12 and *VARS2* exon 17, we showed that the exon inclusion correlated much better with the predicted SRSF1 affinity than 3′ss quality, which were assessed using the Catalog of Inferred Sequence Binding Preferences of RNA binding proteins (CISBP-RNA) database and maximum entropy algorithm (MaxEnt) predictor and the U2AF35 consensus matrix, respectively. RNA affinity purification proved SRSF1 binding to the model 3′ss. On the other hand, knockdown experiments revealed that U2AF35 also plays a role in these exons’ inclusion. Most probably, both factors stochastically bind the 3′ss, supporting exon recognition, more apparently in *VARS2* exon 17. Identifying splicing activators as 3′ss recognition factors is crucial for both a basic understanding of splicing regulation and human genetic diagnostics when assessing variants’ effects on splicing.

## 1. Introduction

Precursor messenger RNA (pre-mRNA) processing (5′ capping, splicing, and 3′ polyadenylation) is a fundamental step in eukaryotic gene expression. Among them, pre-mRNA splicing is a complex process by which introns are removed from pre-mRNA and flanking exons are joined together to form mature mRNA. Pre-mRNA splicing is mediated by the spliceosome, which consists of five small nuclear ribonucleoproteins (snRNPs)—U1, U2, U4, U5, and U6—and a large number of non-snRNP proteins [1,2].

Several sequences are necessary for the spliceosome to recognize splice sites, such as the 5′ splice site (5′ss) and 3′ splice site (3′ss) at the exon-intron borders, as well as the branch point sequence (BPS) and the polypyrimidine tract (PPT) located upstream of the 3′ss. The regulation of both constitutive and alternative splicing involves cis-acting elements, such as trans-acting factors and exonic/intronic splicing enhancers and silencers (ESE/ISE and ESS/ISS, respectively) [3,4,5]. 

Spliceosome assembly takes place in a stepwise manner on each intron. U1 snRNP binds to the 5′ss, splicing factor 1 (SF1) recognizes the BPS and the U2 snRNP auxiliary factor (U2AF) is responsible for recognition of 3′ss [6]. The 65-kDa subunit of the U2AF heterodimer (U2AF65) binds to the PPT, and the 35-kDa subunit (U2AF35) contacts the conserved AG dinucleotide at the 3′ss [7,8]. Then, the U2AF complex recruits U2 snRNP, which replaces SF1 bound to the BPS [8]. In addition, other trans-acting factors, such as SR proteins, are included in the spliceosome assembly to facilitate contacts between U2AF35 and U1 snRNP across the exon [9,10].

SR proteins are involved in the regulation of pre-mRNA splicing, mostly by their binding to ESE and ISE elements [11,12], but they can occasionally bind to silencers as well [13,14]. SRSF1 is one of the best-characterized SR proteins that regulates multiple biological pathways; it plays a role in genomic stability, mRNA transcription, nuclear export, nonsense-mediated mRNA decay, and translation [6,15,16]. SRSF1 regulates both constitutive and alternative splicing [17] by facilitating 5′ss selection and helping to bridge 5′ and 3′ss [9,10,18]. Though SR proteins can often substitute each other in their biochemical roles, the SRSF1 is necessary both during embryogenesis and for genomic stability [19]. 

In exon recognition, the U2AF35 subunit has two functions: (i) it increases the U2AF heterodimer’s RNA binding capacity compared to U2AF65 alone, and (ii) it enlarges the surface for interaction with other splicing factors, such as SR proteins [20]. However, only a fraction of exons seems to be highly dependent on U2AF35, as its knockdown has been shown to only affect a small exome portion [21]. Fu et al. [22] showed a relationship between splicing affected by U2AF35 knockdown and AG-dependence: AG-dependent 3′ss (generally those with a shorter PPT) were shown to rely on U2AF35 for U2AF binding to RNA and on an intact first exonic nucleotide (E+1) for proper splicing. The opposite dependence was demonstrated for AG-independent 3′ss [22]. However, further experiments demonstrated that there is a high variability among the AG-dependent and -independent 3′ss, both in terms of the 3′ss sequence and the extent of the splicing aberration caused by limited U2AF35 concentrations [22,23], thus suggesting that the AG dependence is not a simple binary variable.

During recent years, several alternative ways of 3′ss recognition have been proposed. Above all, both U2AF65- and U2AF35-homologues (U2AF26, Urp, and PUF60) have been described, some of which are capable of forming a functional heterodimer that is both able to bind the 3′ss and mediate SR protein interactions [24]. In line with that, 12% of exons have been shown to be U2AF-independent [25]. It is of note that even U2AF recognition may be further regulated by other proteins with affinity to the PPT that may either support or compete with U2AF binding (e.g., CELF, Sam68, and YB1; and hnRNP C, TDP43, PTB, respectively) [26,27].

Among all this heterogeneity, several 3′ss have been proposed to contain an ESE sequence, a notion supported either by prediction [23,28] or by in vitro experiments [29]. However, it remains unclear whether that an ESE may be functional when located right at the intron–exon border, which has not been unambiguously demonstrated yet.

In this study, we demonstrate for the first time that some 3′ss recognition could be dependent on splicing regulatory proteins binding to an ESE spanning the 5′ end of exons. To inspect this mechanism, we analyzed two model exons (*BRCA2* exon 12 and *VARS2* exon 17) with a GGAGAA SRSF1 binding sequence located at the intron–exon border. Our results suggest that the 5′ end of our model exons includes an enhancer sequence and SRSF1 is one of its recognition factors.

## 2. Results

### 2.1. GGAGAA Sequence Acts as a Splicing Enhancer 

The aim of this research was to determine whether the 3′ss can be recognized through splicing stimulatory proteins binding to an ESE sequence located at the intron–exon border. For that purpose, we chose one of the well-characterized SRSF1 binding motifs, 5′-GGAGAA-3′ [30,31]. An analysis of genomic data showed that the GGAGAA motif located across 3′ss is rather frequent in the human genome. Of the total 188,539 exons (protein-coding exons), 701 exons (0.37%) contained this motif when starting at the −1 position and 262 exons (0.14%) when starting at the +1 position, representing the 99th and 100th percentile among all hexamer motifs at these locations, respectively (Appendix A).

To confirm that the GGAGAA sequence acts as a splicing enhancer element, we employed an ESE-dependent splicing assay [32,33]. This method is based on a minigene assay in which the inclusion of the middle exon depends on whether it contains the ESE. As a reference sequence, we chose a 30 bp fragment containing five TATGGC sequence motifs (Figure 1A). This reference sequence, designated as an NEC (no ESE control), was previously shown to be splicing neutral [34] because it did not create any overlapping binding sites for splicing factors inside TATGGC motif nor at the border of two adjacent motifs as predicted by SpliceAid2 [35]. To analyze the GGAGAA sequence’s effect, the middle TATGGC motif was substituted for the GGAGAA sequence, and the created construct was designed as a putESE (putative ESE) (Figure 1A). 

The results of this minigene assay showed that the GGAGAA sequence is able to induce almost total middle exon inclusion compared to the null inclusion observed for the reference sequence (Figure 1B,C). This is consistent with multiple in silico analyses that demonstrated that this sequence may stimulate splicing and potentially bind the splicing stimulatory proteins (Figure 1D). Taken together, these results confirmed that the GGAGAA sequence has enhancing properties.

### 2.2. GGAGAA Mutations at the Intron–Exon Border Affect Splicing in a Way That Does not Unambiguously Correlate with the Predicted 3′ss Strength

Next, we aimed to examine whether the GGAGAA sequence located at the intron–exon border really acts as an ESE. From the exons containing the desired motif located at the intron–exon border, we selected *BRCA2* exon 12 and *VARS2* exon 17 as model exons. We generated minigene constructs containing the analyzed exons with their flanking intronic sequences and prepared several mutant minigene variants (Figure 2A,C). The mutations were chosen as disrupting the potential ESE and having various effects on the 3′ss strength as estimated by the maximum entropy algorithm (MaxEnt) splice site predictor (Figure 2B,D) [36]. Mutant variants affected both exons’ splicing (exon 12 of *BRCA2* and exon 17 of *VARS2*) (Figure 2B,D). In order to distinguish the role of potential ESE disruption from 3′ss disruption (i.e., presumably impairing U2AF35′s binding capacity), we compared the minigene assay results with the predicted 3′ss strength (MaxEnt score) (Figure 2) and the predicted similarity to the U2AF35 consensus binding site (using scores proposed by Doktor et al. [37]) (Appendix A). *BRCA2* exon 12 inclusion was moderately correlated to both the MaxEnt score (r = 0.700 and *p* = 0.053) (Figure 2B) and the U2AF35 score (r = 0.685 and *p* = 0.061) (Appendix A), though on the border of statistical significance. On the other hand, *VARS2* exon 17 inclusion showed no evident correlation to any of these scores (r = 0.454 and *p* = 0.259 for MaxEnt score and r = 0.286 and *p* = 0.492 for U2AF35 score) (Figure 2D and Appendix A). The discrepancy between splicing affection and the correspondence to the splice site consensus was nicely illustrated by exon inclusion in E+3T and E+4T *BRCA2* minigene construct variants when compared to the wild-type constructs. While these two variants did not decrease the MaxEnt score, they both led to significantly decreased exon inclusion (Figure 2B). Similarly, in *VARS2*, the E+2T variant decreased exon inclusion compared to the wild-type construct while increasing the 3′ss quality, as estimated by both the MaxEnt and U2AF35 scores (Figure 2D).

Rather unsurprisingly, comparing these predictions suggested that the 3′ss exonic mutants’ MaxEnt score may even reflect the U2AF35 binding preferences (Appendix A) [7]. In support of that notion, we compared the predicted MaxEnt and U2AF35 scores for *BRCA2* and *VARS2*-derived sequences corresponding to our model 3′ss (but mutated to all possible variants in the exonic part) and found a highly significant correlation between these two scores (r = 0.658 and *p* = 3.4 × 10^−9^ for *BRCA2*; r = 0.573 and *p* = 7.5 × 10^−7^ for *VARS2*) (Appendix A; Appendix A).

To assess the extent to which SRSF1 binding could be relevant for model 3′ss recognition, we consulted the Catalog of Inferred Sequence Binding Preferences of RNA binding proteins (RBPs) (CISBP-RNA) database [31], which assigns a Z-score as a measure of individual RBP binding affinity. Using SRSF1-derived Z-scores, we observed a significant correlation between heptamers’ Z-scores spanning our model constructs and their exon inclusion (Figure 3, Appendix A). Naturally, the predicted Z-scores varied according to the nucleotide selected as the heptamer’s first position. In *BRCA2* exon 12, the correlation between exon inclusion and Z-score for putative SRSF1 binding sites was 0.912 (*p* = 0.002) when starting at the −1 position (Figure 3A and Appendix A), and the correlation was 0.873 (*p* = 0.005) when starting at the −2 position (Appendix A). In *VARS2* exon 17, analyzing 7-mers corresponding to individual RBPs showed that the Z-score and exon inclusion correlation for SRSF1 binding sites was 0.773 (*p* = 0.024) when starting at the −4 position (Figure 3B and Appendix A) and 0.663 (*p* = 0.073) when starting at the −1 position (Appendix A). As there is no research showing effects of SRSF1 precise localization at 3′ss, we assume that SRSF1 might theoretically bind anywhere at the intron–exon border. Therefore, based on all the Z-score predictions starting at individual positions between −6 and +1, we used the maximal correlation coefficient in this manuscript. The complete list of the Z-scores at individual starting positions and their Pearson’s correlation coefficients with exon inclusion can be found in the Appendix A (Appendix A).

Concerning the peculiar mutants mentioned above, the exon inclusion changes in E+3T and E+4T variants in *BRCA2* and in the E+2T variant in *VARS2* conformed much better to Z-score changes than to the MaxEnt or U2AF35 scores (Figure 2 and Figure 3, Appendix A; Appendix A). Overall, these results suggested that apart from the 3′ss quality, as scored by MaxEnt, the GGAGAA motif at the intron–exon border may contain a potential enhancer that the SRSF1 protein is able to recognize (Figure 3) and that may promote exon inclusion.

### 2.3. SRSF1 and U2AF1 Are Included in Recognition of Both Exons

To assess the role of SRSF1 as a positive regulator for our model exons, we performed an overexpression experiment in the HeLa cell line. The results showed that the SRSF1 overexpression markedly inhibited exon inclusion in *BRCA2* exon 12 (Figure 4A). On the contrary, SRSF1 overexpression strongly promoted *VARS2* exon 17 exon inclusion (Figure 4C). As the increase in exon inclusion was similar in all 3′ss variants except for the wild-type construct (Figure 4C), SRSF1 probably binds elsewhere in the exon to promote splicing, possibly in addition to binding to 3′ss. In addition, given that splicing factors often regulate the expression of other splicing regulators [21,38], we have to consider the possibility that we observed secondary effects stemming from other splicing regulators’ expression changes rather than those of SRSF1.

To detect whether and to what extent U2AF35 could control exon inclusion through 3′ss recognition in our models, we tested the effect of U2AF35 overexpression (Figure 4B,D). In fact, we did not detect any effect of U2AF35 overexpression on our model exons inclusion except for the *VARS2* E+2T variants (Figure 4D). A possible explanation for this is that the U2AF35 cellular level may be high enough to ensure maximal U2AF35 RNA saturation.

To further confirm the role of SRSF1 and U2AF35 on our model exon recognition, these proteins were silenced in HeLa cells by siRNA-mediated knockdown (Figure 5). The siRNA-mediated knockdown of SRSF1 caused a rather slight (statistically non-significant) and uniform decrease in *BRCA2* exon 12 inclusion in the wild-type construct and all analyzed variants (Figure 5A). The same pattern was observed for *VARS2* exon 17 (Figure 5C). U2AF35 knockdown led to a strong decrease in exon inclusion in the wild-type construct and the 2G variants, as well as a moderate effect in other variants for *BRCA2* exon 12 (Figure 5B). In *VARS2* exon 17, U2AF35 knockdown caused a marked decrease in exon inclusion in all variants except for the variants at the first position of the exon (Figure 5D). Based on these results, it seems that the effect of U2AF35 knockdown is more pronounced than effect of SRSF1 knockdown. The results suggested that both SRSF1 and U2AF35 could be included in the recognition of these exons. However, the similarity in SRSF1 knockdown effect in all the variants indicated that this protein might bind to somewhere other than to the tested 3′ss, either exclusively or in addition to the 3′ss binding.

### 2.4. GGAGAA Motif Can Be Recognized by SRSF1 in Our Models

To further demonstrate whether ESE mutations lead to abrogated SRSF1 binding in our model 3′ss, we performed the affinity purification of RNA-binding proteins. As a control of specific binding to our 3′ss, we compared target protein binding to wild-type and mutated probes. The E+1T variant was selected for its markedly lower SRSF1 affinity Z-score [31] than the that of the wild-type construct (Figure 6A and Appendix A). In addition, we tested a double mutant E+1T2T that should have completely abolished the SRSF1 binding site (Figure 6A and Appendix A) [31].

The wild-type and both mutant RNA oligonucleotides were used in a pull-down experiment with a HeLa nuclear extract, followed by Western blot analysis. In *BRCA2* exon 12, there was a lesser SRSF1 signal in the E+1T2T mutant variant compared to the wild-type construct (Figure 6B), while the effect of E+1T mutation varied between individual experimental repetitions (data not shown). Therefore, the SRSF1 binding at this site is possible but not convincingly proven. On the other hand, both *VARS2* mutant variants resulted in decreased SRSF1 binding when compared to the wild-type construct (Figure 6B). These data indicated that the 5′ end of *VARS2* exon 17 probably contains the binding site for SRSF1. In addition, other splicing regulators were observed to bind differently to our wild-type and mutant probes; for instance SRSF5 binding was decreased in both *BRCA2* variants compared to the wild-type construct, but it was increased in *VARS2* mutants compared to the wild-type construct (Figure 6B and Appendix A).

The affinity purification of RBP was used to identify RNA–protein interactions, but it happened that indirect protein–protein binding to RNA was detected as well. In order to rule out the idea that the observed change in SRSF1 binding may have been secondary due to its interaction with U2AF35, we also tested the U2AF35 binding. We observed both a decrease and increase in U2AF35 binding in individual experimental repetitions with no correlation to the SRSF1 binding. The data suggested that SRSF1 did not bind to our exons through interaction with U2AF35. The results showed that the GGAGAA sequence motif can be bound by SRSF1 in both *BRCA2* exon 12 and *VARS2* exon 17. In addition, other factors could affect these sequences, as well as exon splicing.

## 3. Discussion

During the first spliceosome assembly steps, most acceptor splice sites are recognized by the U2AF heterodimer. Binding its larger subunit (U2AF65) to the PPT is often facilitated by a smaller subunit (U2AF35) that binds the 3′ss AG (at least at AG-dependent 3′ss) and/or mediates exon recognition by interacting with SR proteins and other splicing activators [7,26]. Surprisingly, a rather limited number of exons are strictly dependent on U2AF35 [21], and a significant portion (about 12%) of exons may not be dependent on U2AF heterodimer binding or function at all [25]. In many of these exons, the intron–exon border (ag/N) may be available for binding of other splicing factors that can either interact directly with U2AF65 or U2 snRNP. Alternatively, splicing activators bound to the intron–exon border may interact with U2AF35 (itself not binding the RNA), thus attracting the U2AF65 to the PPT and helping recognize the 3′ss.

In this work, we tested the hypothesis that an ESE recognized by SR proteins can be located at the intron–exon border. Such a case has been already proposed by Rooke et al. [29] for the *c-src* gene’s N1-exon where SRSF1 binds to the 3′ss. However, the results did not clearly demonstrate whether the SRSF1 binding site stretched to the ag/N or started a few nucleotides downstream. In any case, simultaneous SRSF1 and U2AF35 binding to this sequence is highly improbable, since the U2AF35 binding site has been shown to span at least 10 exonic nucleotides [7]. However, functional ESEs located even in the first ten exonic nucleotides may not be rare [41,42,43,44,45].

Generally, SRSF1 can promote or prevent exon recognition in a context-dependent manner, depending on the sequence to which they bind [46,47], and it tends to recognize purine-rich RNA sequences. As involved in both 5′ss and 3′ss recognition, SRSF1 interacts with U1-70K, a component of U1 snRNP, to recruit U1 snRNP to 5′ss and with U2AF35 to assist 3′ss recognition [9,10]. However, many other factors can interact with SRSF1, including U2AF65 [48,49].

Interestingly, the minigene assay for the two model exons showed that the effect of GGAGAA motif mutations on splicing correlated much better with the predicted binding affinity of SRSF1 (Figure 3, Appendix A) than with the overall 3′ss strength (Figure 2) or U2AF35 consensus (Appendix A) [31,37], which is in line with an ESE located at the 5′ end of exons.

Using an ESE-dependent splicing assay, we confirmed that the GGAGAA motif has enhancing properties when inserted in the middle of an ESE-dependent exon (Figure 1). In contrast, the GGAGAA motif’s enhancer function was not conclusive when this motif was located at the 5′ end of an artificial ESE-dependent exon; multiple motif mutations showed bigger effects of 3′ss strength on splicing compared to the ESE disruption (data not shown), which may have stemmed from the highly artificial nature of this artificial, just 34-nt long, exon.

Interestingly, our knockdown experiments demonstrated that the model exons depended more on the U2AF35 level than the SRSF1 level (Figure 5), which was quite unexpected regarding the results of mutated minigenes analyses related to binding site disruptions. SRSF1 might be just one of several recognition factors in the studied 3′ss sequences. As SREs often tend to overlap [50], the 3′ss might be recognized by a set of splicing regulators (in a rather stochastic manner), many of them possibly interacting with U2AF35. Using the oRNAment database [51], which is able to map potential RBP binding sites in the coding RNA/transcriptome, the SRSF1 binding site (GGAGAAG) was found to be located at 5′ end of *VARS2* exon 17, while only other splicing factors binding sites than that of SRSF1 were predicted for *BRCA2* exon 12 (Appendix A). In silico predictions for our 3′ss further identified potential binding sites for SRSF5 [35], SRSF9 (Appendix A) [31], Tra2-β1 [52], 9G8, hnRNP A1 [53], and SRSF2 [17]. Alternatively, the larger effect of U2AF35 knockdown compared to SRSF1 knockdown may represent a secondary effect that reflects the interconnection between individual splicing factor expressions. It has been well-documented that splicing factors often regulate several other splicing regulators’ expressions, even their own interaction partners [21,38]. In that case, the effect of the knockdown might reflect the general changes in the expression of the splicing regulators that play roles in our exons’ recognition rather than their direct binding to 3′ss. Because we used the cancerous cell line HeLa in our experimental procedures, we had to count with the inherent splicing factors’ deregulated expression, which is a common phenomenon in many cancer types [54]. Therefore, even the intrinsic SRSF1 and U2AF35 expression in our cells may differ from non-malignant mammalian tissues. However, we believe that both these factors are sufficiently expressed and functional in the HeLa cells because this cell line has been repeatedly used for research on these factors, as well as splicing in general [55,56,57,58,59].

In line with that notion, individual mutations did not markedly vary in the splicing response to the SRSF1 knockdown. This may have been related to requirement of multiple SRSF1 binding sites to ensure the knockdown’s function, as described by Jobbins et al. [60]. Therefore, a single nucleotide substitution might not have significantly harmed the SRSF1 binding. Indeed, searching for SRSF1 binding motifs showed multiple potential binding sites in both of our model exons (Appendix A).

In any case, the marked effect of a U2AF35 limitation on splicing, together with sequence-specific splicing response in *VARS2*, showed that this factor is important for our model exons’ recognition (Figure 5D). This can be explained by two possible models: Either the 3′ss ag/N is stochastically bound by U2AF35 or SRSF1 (or other factors), both of which promote its use (as proposed in Goren et al. [50]), or the SRSF1 and/or possibly other factors bind the intron–exon border and attract U2AF65 by interacting with U2AF35. In *VARS2* exon 17, the former mechanism offers a more probable explanation to the obtained data. In fact, both the exons’ inclusion in response to the U2AF35 knockdown (Figure 5D) correlated well with the U2AF35 motif score (Appendix A), which suggested that U2AF35 is the recognition factor for this 3′ss. However, the affinity purification analysis clearly also showed the SRSF1 binding to the 3′ss sequence (Figure 6B), which was in line with multiple predictions (Figure 3B, Appendix A) and the effects of mutations observed in the minigene analysis (Figure 2).

On the other hand, in the case of *BRCA2*, our results could not well-distinguish between the two mechanisms proposed above. SRSF1 binding to this 3′ss was both predicted and suggested by splicing minigene assay results (Figure 2B) and their correlation to SRSF1 affinity predictions (Figure 3A, Appendix A). In addition, affinity purification confirmed SRSF1 binding to this site, even though it was less clearly disrupted by the 3′ss mutations than in the *VARS2* case in some experimental repetitions (Figure 6). However, the effect of U2AF35 knockdown was rather uniform across diverse 3′ss mutants (Figure 5B) and did not correlate clearly with the U2AF35 score (Appendix A), which might, in fact, have arisen from the lower number of analyzed variants. We still believe that several other factors may bind the inspected 3′ss in a rather stochastic manner, as suggested by predictions and affinity purification results (Figure 6 and Appendix A).

Despite the inherent limitations of the affinity purification assay, we consider SRSF1 binding to the intron–exon border to be rather direct and specific because (i) no other SRSF1 binding site was predicted in the tested probes, (ii) mutations disrupting the consensus binding motif reduced the SRSF1 signal, (iii) undirect pull-down due to SRSF1 interaction with U2AF was not probable because we did not include a U2AF65 binding site (the PPT) in our probes and the U2AF35 binding to RNA was not correlated to SRSF1 binding. Still, we cannot exclude the possibility of SRSF1 binding due to interaction with splicing factors other than U2AF35, but we consider that unlikely because the reduced SRSF1 signal in the variant that disrupted its consensus binding motif (1T2T) strongly suggested that SRSF1 could directly bind to our model exons. Because of the lack of a binding site for its partner, U2AF65, in the probes, we would not like to infer any conclusion about the U2AF35 binding from the affinity purification assays.

Overall, the results indicated that our model 3′splice sites contain an ESE right at the intron–exon border that can be recognized by SRSF1 and possibly other splicing regulators. Despite both SRSF1 and U2AF35 being included in the recognition of our model exons, the data suggested that U2AF35 may not have directly bound to our 3′ss, at least in the case of *BRCA2*.

Based on these findings, we assume that the *BRCA2* exon 12 3′ss is recognized by SRSF1 binding that may interact with U2AF35 (itself not binding the RNA), thus attracting the U2AF heterodimer to this site. Still, we do not rule out other splicing factors’ participation, which may compete with SRSF1 for the 3′ss binding. On the other hand, *VARS2* exon 17 recognition conformed better to a model in which U2AF35 and SRSF1 compete and stochastically bind the 3′ss sequence, both of them supporting splicing at this site. Again, the participation of other splicing factors is possible. Therefore, we propose a mechanism of 3′ss recognition that would allow for the bypass of the U2AF35 binding yet preserving its need during the 3′ss recognition. We hope that these results may contribute to a better understanding of the 3′ss recognition mechanism and possibly to more reliable predictions of splice site mutations’ effects.

## 4. Materials and Methods

### 4.1. In Silico Prediction/Bioinformatic Analysis

MaxEntScan (http://hollywood.mit.edu/burgelab/maxent/Xmaxentscan_scoreseq_acc.html) [36] was used to predict splice site strengths. To predict the effect of hexamers, the ESRseq score described by Ke et al. [34] and the Z_EI_ scores derived from Erkelenz et al. [61] were used. From the CIS-BP RNA database (http://cisbp-rna.ccbr.utoronto.ca) [31], the Z-score of the chosen RBPs for all 7-mers spanning individual variants were obtained and compared with each other. Putative RBP binding sites were predicted using the Human Splicing Finder (http://www.umd.be/HSF/technicaltips.html) [53], oRNAment (http://rnabiology.ircm.qc.ca/oRNAment/) [51], and SpliceAid2 (http://193.206.120.249/splicing_tissue.html) [35], all of which enabled the prediction and mapping of known splicing factors’ binding sites.

### 4.2. Splicing Minigene Assay

#### 4.2.1. Plasmid Construction and Mutagenesis

Exons with part of their flanking introns (Appendix A) were amplified using Platinum Pfx DNA polymerase (Life Technologies, Carlsbad, CA, USA) and primers carrying the *BamHI* and *XhoI* restriction sites (Appendix A) from the DNA of a healthy control. The amplified products were subsequent cloned in a modified version of the pET01 vector (MoBiTec, Göttingen, Germany). The identity of new minigene constructs was checked by sequencing on an ABI PRISM ^®^ 3130 Genetic Analyzer (Thermo Fisher Scientific, Prague, Czech Republic). The PCR conditions are available upon request.

Mutagenesis was carried out with PrimeSTAR Max DNA polymerase (Takara, Shiga, Japan) using specific primers (Appendix A). Wild-type minigenes were used as templates to generate several variants at the 5′ end of each exon. The mutations were verified by sequencing.

#### 4.2.2. Cell Culture and Transfection

The HeLa (obtained from the European collection of cell cultures, ECACC, England, United Kingdom) cell lines were cultured in an RPMI 1640 medium supplemented with L-glutamine (Sigma Aldrich, Prague, Czech Republic) and 10% fetal bovine serum (FBS; Sigma Aldrich, Prague, Czech Republic), and then they were incubated at 37 °C in a humidified 5% CO_2_ condition.

HeLa cells were seeded into a 12-well plate to achieve a 30% confluency in 24 h. For the splicing minigene assay, the cells were transfected with 800 ng of DNA using the X-treme Gene 9 transfection reagent (Roche, Prague, Czech Republic) according to the manufacturer’s instructions.

For the overexpression experiment, HeLa cells were co-transfected with 800 ng of our minigene plasmids and 800 ng of the overexpression plasmids (pcDNA-FLAG_SF2, which was a gift from Honglin Chen (Addgene plasmid #99021; http://n2t.net/addgene:99021; RRID:Addgene_99021, Watertown, MA, USA) [39], or pRRL_U2AF1_WT_mCherry, which was a gift from Robert Bradley (Addgene plasmid #84017; http://n2t.net/addgene:84017; RRID:Addgene_84017, Watertown, MA, USA [40]) using the X-treme Gene 9 transfection reagent according to the manufacturer’s instructions. Empty expression vectors were used as a control.

For siRNA transfection, MISSION^®^ esiRNA purchased from Sigma Aldrich was used. HeLa cells seeded in 12-well plates were transfected with 2.5 uL of Lipofectamine^TM^ RNAiMAX Transfection Reagent (Thermo Fisher Scientific, Waltham, MA, USA) and 10 uM siRNA targeting SRSF1 (Sigma Aldrich, Prague, Czech Republic) or U2AF35 (Sigma Aldrich, Prague, Czech Republic) in an OptiMEM-I Reduced Serum Medium (Thermo Fisher Scientific, Waltham, MA, USA). siRNA targeting Upf1 (Thermo Fisher Scientific, Waltham, MA, USA) was used as a control. After 48 h of incubation, the medium was changed to the RPMI-1640 supplemented with 10% FBS and transfection with 500 ng of our constructs, and the X-treme Gene 9 transfection reagent was used according to the manufacturer’s instructions. Transfections were performed in three independent biological replicates.

#### 4.2.3. RNA Extraction and RT-PCR

Twenty-four hours after DNA transfection, total RNA was extracted using the Quick-RNA™ MiniPrep Kit (Amplicon, Prague, Czech Republic) according to the manufacturer’s instructions. The concentration and purity of the RNA samples were assessed using Nanodrop^®^ 1000 (Thermo Fisher Scientific, Erembodegem, Belgium). The reverse transcription was performed using 200 ng of total RNA and the Transcriptor First Strand cDNA Synthesis Kit (Roche, Prague, Czech Republic) primed with random hexanucleotides according to the manufacturer’s instructions.

To analyze the splicing pattern, RT-PCR was performed to determine the exponential phase of PCR amplification using Taq DNA Polymerase (Thermo Fisher Scientific), Eva Green Dye (Biotium, CA, USA) and primers pET_f, and pET_r (Appendix A). RT-PCR quantitation was carried out on Rotor-Gene Q (Applied Biosystems, CA, USA) and the Rotor-Gene G Series Software (Applied Biosystems, CA, USA). The PCR conditions are available upon request.

#### 4.2.4. Capillary Electrophoresis Analysis

To quantify splicing isoforms, capillary electrophoresis was used. To preserve the exponential PCR phase, cDNA samples were diluted to unify the cDNA concentration and then amplified with Taq DNA Polymerase and the primers pET_FAM_f and pET_r (Appendix A). PCR was run for the same number of cycles. Then, four percent of each PCR product was mixed with 9 uL of Hi-Di Formamide (Thermo Fisher Scientific, Waltham, MA, USA) and 0.25 uL of GeneTrace 1000^TM^ Standard (Carolina Biosystems, Prague, Czech Republic), which was denatured and frozen. Samples were run on the ABI PRISM^®^ 3130 Genetic Analyzer, and the obtained data were analyzed by the GeneMapper 4.1 (Thermo Fisher Scientific, Prague, Czech Republic). Mean peak areas were used to calculate the different splicing transcript ratios generated by the minigene variants. The results from independent biological replicates are expressed as mean ± SD. The statistical analysis of RT-PCR data was carried out using a two-tailed *t*-test (Student’s unpaired) (Statistica 12 software, StatSoft, Tulsa, OK, USA) with a *p*-value cut-off of 0.05.

### 4.3. Quantitative PCR

To detect overexpression and siRNA-mediated knockdown efficiency, SRSF1 and U2AF35 expression levels were measured using TaqMan Gene Expression Assays (Thermo Fisher Scientific, Foster City, CA, USA) on a LightCycler 480 Instrument II (Roche, Prague, Czech Republic) (hs01044658_g1, hs00739599_m1; Thermo Fisher Scientific). Obtained data were analyzed in the LightCycler 480 SW 1.5.1 (Roche, Prague, Czech Republic) using the fit points method and normalized to GAPDH housekeeping gene’s expression (hs99999905_m1; Thermo Fisher Scientific, Foster City, CA, USA). The upregulated mRNA levels were verified in selected variants.

### 4.4. ESE-Dependent Splicing Minigene Assay

The GGAGAA motif was tested for the presence of an ESE using an ESE-dependent splicing assay [32,33]. The tested sequence was obtained by annealing complementary oligonucleotides carrying 5′-*EcoRI* and 3′-*BamHI* compatibles ends (Appendix A) and inserted into the *EcoRI* and *BamHI* restriction sites of the pcDNA-Dup plasmid. Control samples—including a negative control (a pcDNA-Dup-BRCA2int11 construct, which contains a fragment without known regulatory elements) and a positive control (a pcDNA-Dup-SC35 construct containing a triplet of binding sites for the SR protein SC35)—were tested.

To analyze the tested sequences’ effects, HeLa cells were transfected as described above. After 24 h, total RNA was extracted and reverse transcription was performed as described above. The cDNA was amplified with Taq DNA Polymerase (Top Bio, Vestec, Czech Republic), Eva Green Dye, and the primers T7pro and DupS4_Seq3R (Appendix A). The PCR was carried out on Rotor-Gene Q. The PCR conditions are available upon request. RT-PCR products were separated on the Midori Green Advance 2% agarose gel stain. To semiquantify splicing isoforms, capillary electrophoresis was performed using Taq DNA Polymerase (Top Bio, Vestec, Czech Republic) and the primers T7pro-FAM and DupS4_Seq3R (Appendix A), as described above. The impact of tested sequences, together with positive and negative controls, on middle exon splicing was obtained from three independent repetitions.

### 4.5. Affinity Purification of RNA-Binding Protein (Pull-Down Assay)

To verify whether the SRSF1 or U2AF35 bind tested exons, a pull-down assay was used. We designed wild-type, E+1T, and E+1T2T mutant DNA oligonucleotides comprising 17 nucleotides in the sequence of interest spanning the 3′ss, five nucleotides from the intron, and twelve nucleotides from the exon. Complementary DNA oligonucleotides contained sequences of interest and carried a T7 polymerase target sequence (5′-TAATACGACTCACTATAGGGTAGG-3′) at the 5′ end of the forward primer. DNA templates were obtained by annealing the sequences of interest and used as templates for T7 RNA polymerase (New England Biolabs, Ipswich, MA, USA) driven transcription in vitro. Subsequently, DNA oligonucleotides were degraded by RNase free DNAse I (New England Biolabs, Ipswich, MA, USA) and then purified by phenol/chloroform; this was followed by ethanol precipitation and resuspension in 40 uL of ultrapure water. The affinity purification of RNA-binding proteins was performed as previously reported [62].

### 4.6. SDS-PAGE and Western Blot Analysis

The protein samples were dissolved in an SDS-sample buffer and denaturated for 10 min at 95 °C before loading on a 12% SDS-polyacrylamide denaturating gel (SDS-PAGE). Proteins were visualized by Coomassie Brilliant Blue R (Sigma Aldrich, Prague, Czech Republic) staining and blotted onto a TransBlot^®^ Transfer medium pure nitrocellulose membrane (Biorad, Hercules, CA, USA). In the case of mouse antibodies, membranes were blocked using a 1× Western blocking reagent solution (Roche, Prague, Czech Republic) according to the manufacturer’s instructions. Mouse antibodies (anti-SF2/ASF (96), 1:1000, sc-33652, Santa Cruz Biotechnology, Inc. CA, USA; anti-SR proteins, 1:1000, 33-9400, Life Technologies, Carlsbad, CA, USA) were diluted in a 1× Western blocking reagent solution. For rabbit antibodies, membranes were blocked with 4% non-fat dry milk (Biorad, Hercules, CA, USA) in 1× PBS and 0.1% Tween 20. Rabbit antibodies against U2AF35 (1:1000, 10334-1-AP, ProteinTech, IL, USA) were prepared using 1× PBS and 0.1% Tween 20. The transferred membrane was blocked for 1 h at RT and incubated with primary antibodies overnight at 4 °C. After several washes with Tris-buffered saline containing 0.1% Tween 20 or a PBS buffer containing 0.1% Tween 20, the membranes were incubated with the corresponding secondary antibodies (goat anti-rabbit antibody, 1:2500, 170-5046, Bio-Rad, Hercules, CA, USA; goat anti-mouse secondary antibody, 1:2500, M30107, Life Technologies, Carlsbad, CA, USA) for 1 h at room temperature. After several washes, proteins probed with antibodies were detected with Pierce TM ECL (enhance chemiluminescence system) Western Blotting Substrate (Thermo Fisher Scientific, Prague, Czech Republic), according to the manufacturer’s instructions, and analyzed by Alliance (Uvitec, Cambridge, United Kingdom). To semiquantify the protein signal, the GelQuant.NET (version 1.8.2, BiochemLabSolutions.com, San Francisco, CA, USA) software was used, and the signal was normalized to those obtained from Coomassie staining.

## Figures and Tables

**Figure 1 ijms-21-06553-f001:**
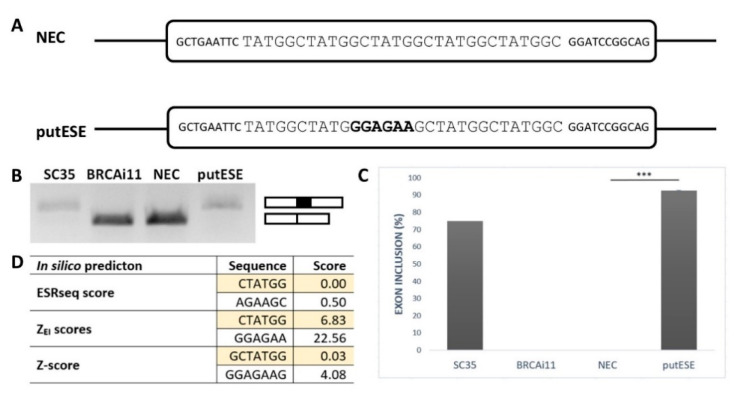
GGAGAA motif acts as a splicing enhancer. (**A**) Schematic representation of no exonic splicing enhancer (ESE) control (NEC) and putative ESE (putESE) sequences inserted into pcDNA-Dup vector middle exon used in an ESE-dependent splicing assay [32,33]. Larger uppercase letters illustrate inserted sequence, and bold letters show the putative ESE motif. (**B**) Agarose gel image shows the RT-PCR analysis of spliced transcripts expressed from NEC and putESE constructs. In parallel, positive controls (SC35) containing three binding sites for SR protein SC35 and negative control (BRCAi11) containing no known regulatory elements derived from *BRCA2* intron 11 were used. (**C**) Inclusion of the middle exon was quantified by capillary electrophoresis, and it is shown as mean ± SD of three independent experiments. Statistically significant difference is shown (*** *p* < 0.001). (**D**) Scores from different prediction tools indicated potential splicing enhancing or silencing properties of analyzed sequences. Enhancer character shown as positive, neutral as zero, and silencer as a negative number; 6-mers/7-mers with the highest score of the overlapping sequence are shown.

**Figure 2 ijms-21-06553-f002:**
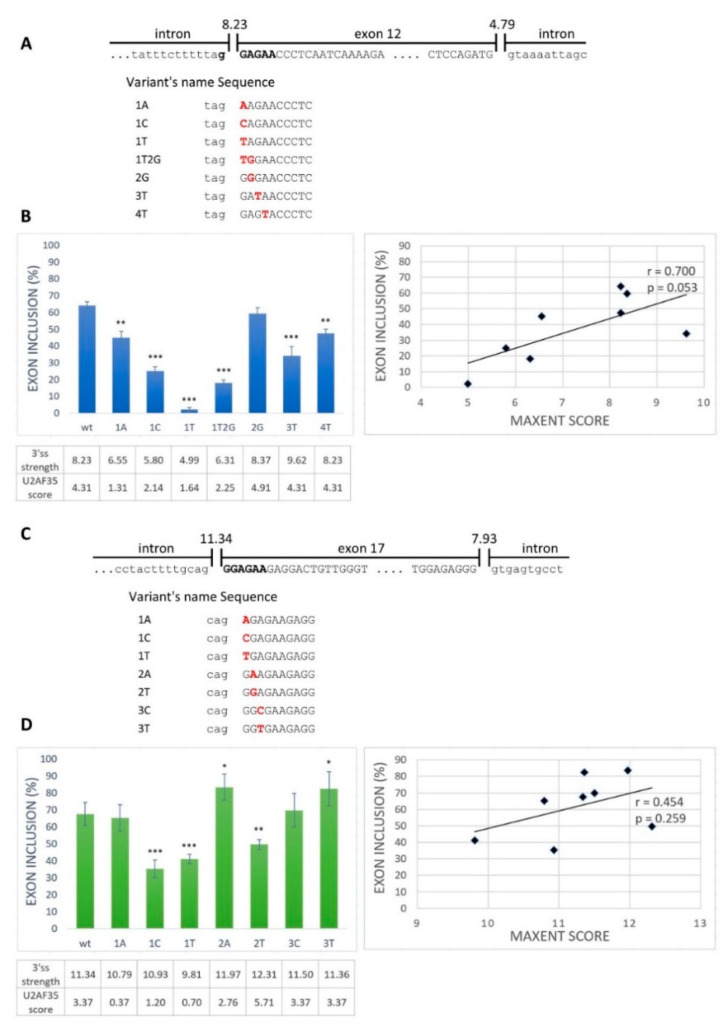
Splicing minigene assay to verify a potential ESE element. (**A**,**C**) Schematic representation of the wild-type sequence and part of flanking introns of *BRCA2* exon 12 (**A**) and *VARS2* exon 17 (**C**). Uppercase letters show exonic bases, and lowercase letters intronic bases. Bold black letters represent the putative ESE binding motif. The designed potential ESE binding motif variants are indicated below. (**B**,**D**) Effects of 3′ splice site (3′ss) mutations on *BRCA2* exon 12 splicing (**B**) and *VARS2* exon 17 splicing (**D**) are shown in graphs. Error bars indicate SD. Statistically significant differences from wild-type inclusion are shown (* *p* < 0.05, ** *p* < 0.01, and *** *p* < 0.001). 3′ss strength (expressed as its max entropy algorithm (MaxEnt) score) and U2 small nuclear ribonucleoprotein (snRNP) auxiliary factor (U2AF35) score (counted according to Doktor et al. [37]) are shown in the table below the graph. On the right, the correlation of the MaxEnt score of 3′ss and the exon inclusion frequency of both *BRCA2* exon 12 (**B**) and *VARS2* exon 17 (**D**) are shown.

**Figure 3 ijms-21-06553-f003:**
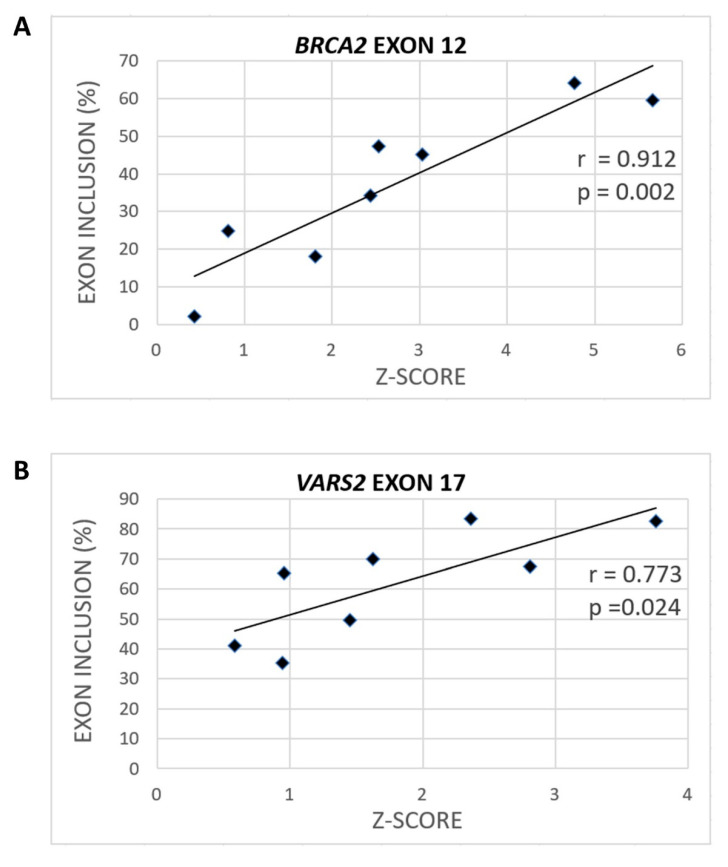
Various variants can have different effects on SRSF1 binding (Z-scores). The maximal correlation coefficient between exon inclusion and potential SRSF1 binding (shown as the Z-score) for *BRCA2* (**A**) when using the SRSF1 binding site starting at the −1 position and *VARS2* (**B**) when starting at the −4 position is shown.

**Figure 4 ijms-21-06553-f004:**
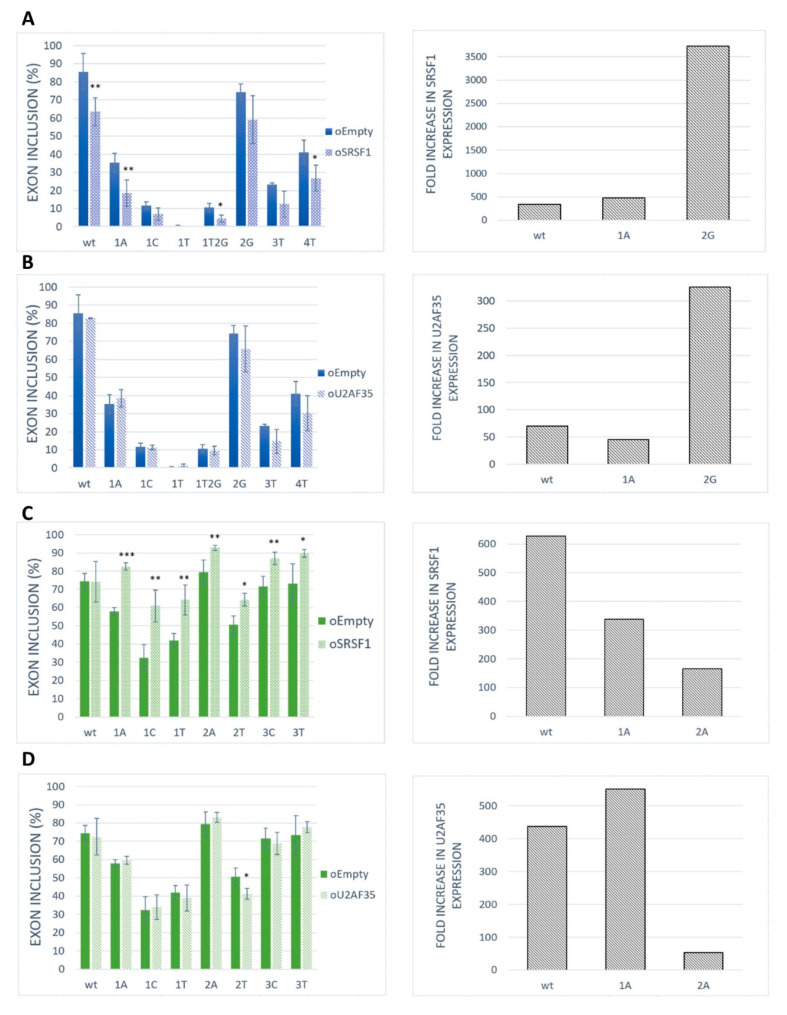
Effect of *SRSF1* and *U2AF35* overexpression on exon recognition. HeLa cells were co-transfected with *SRSF1* [39] or *U2AF35* [40] expression vectors and *BRCA2* or *VARS2* minigenes. An empty vector was used as a control. Graphs show proportions of exon inclusion upon *SRSF1* overexpression (**A**,**C**) or *U2AF35* overexpression (**B**,**D**) (light bars) compared to control (dark bars) in *BRCA2* exon 12 (blue) (**A**,**B**) or *VARS2* exon 17 (green) (**C**,**D**). Data are shown as mean exon inclusion percentage ± SD obtained from three independent experiments. Statistically significant differences are shown (* *p* < 0.05, ** *p* < 0.01, and *** *p* < 0.001). On the right, *SRSF1* or *U2AF35* expression levels are shown as fold-change with respect to control transfections. *SRSF1* or *U2AF35* expression level upon SRSF1 or U2AF35 overexpression relative to SRSF1 or U2AF35 level from control transfection is shown.

**Figure 5 ijms-21-06553-f005:**
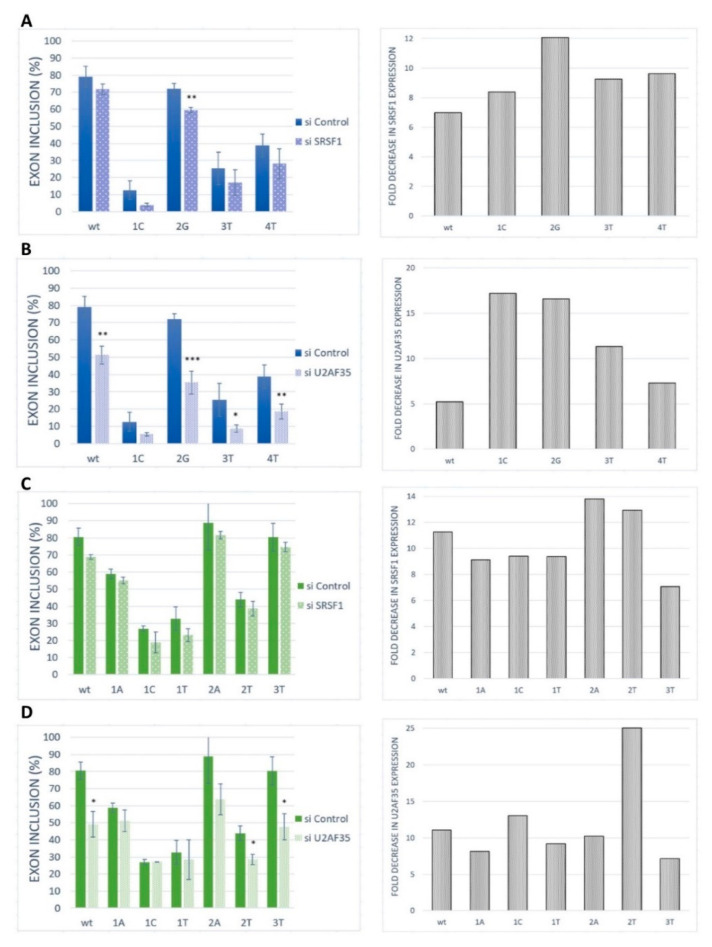
Effect of SRSF1 and U2AF35 knockdown. HeLa cells were transfected with either a control (si control) or specific siRNA (si SRSF1 or si U2AF35). Graphs demonstrate the proportions of exon inclusion upon SRSF1 (**A**,**C**) or U2AF35 (**B**,**D**) knockdown in *BRCA2* exon 12 (**A**,**B**) or *VARS2* exon 17 (**C**,**D**). Exon inclusion from different minigenes is compared to control transfection (si control). Error bars represent SDs obtained from three independent experiments. Statistically significant differences are shown (* *p* < 0.05, ** *p* < 0.01, and *** *p* < 0.001). On the right, *SRSF1* or *U2AF35* expression levels are shown as fold-change compared to control transfections (si control). *SRSF1* or *U2AF35* expression level upon SRSF1 or U2AF35 knockdown (grey bars) relative to *SRSF1* or *U2AF35* level upon control knockdown is shown.

**Figure 6 ijms-21-06553-f006:**
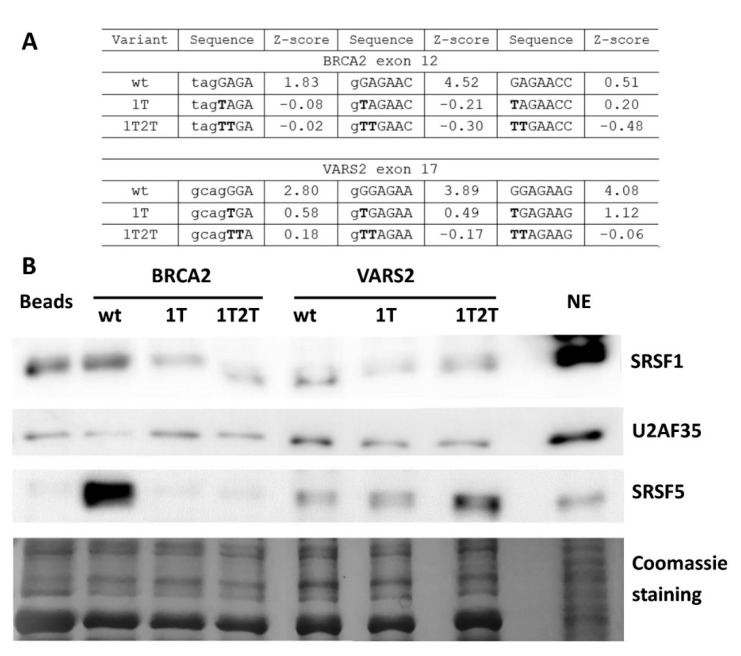
Affinity purification of RNA binding proteins (RBPs) of in vitro transcribed RNAs. (**A**) Table shows SRSF1 binding motif weakening in 1T and 1T2T variants compared to the wild-type construct via their Z-scores. Values correspond to the binding affinity of SRSF1 to each 7-mer [31], and 7-mers with the highest Z-score in analyzed sequences are shown. (**B**) Western blot analysis after the affinity purification of RNA-binding proteins is shown. Beads alone (Beads) and the HeLa nuclear extract (NE) sample were used as a control. Western blot images and Coomassie gel staining are representative figures in three independent experiments.

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
