# Peer review of "Splicing Enhancers at Intron–Exon Borders Participate in Acceptor Splice Sites Recognition"

_ijms, 2020, doi:10.3390/ijms21186553_

Round 1

Reviewer 1 Report

This is a very interesting paper regarding splice site characterization and in particular the 3’ acceptor splice site recognition as a  fundamental step in pre-mRNA splicing. The U2AF  heterodimer recognizes the 3’ ss and it bounds to the exon border mediating the enhancer-binding splicing activator interactions with the splice some. As alternative mechanism  the exon border is bound by the ubiquitous splicing regulator SRSF1.

In this work the au tested the hypothesis that an ESE (GGAGAA motif), recognized by SR proteins, can work when  located at the intron-exon border as already proposed for the c-src gene.

Using minigene assays of two model exons BRCA2 exon 12 and VARS2 exon 17, the au demonstrated that the exon inclusion correlated much better with the predicted SRSF1 affinity than the 3’ ss quality assessed using the CISBP-RNA database and MaxEnt predictor together with the U2AF35 consensus matrix, respectively. RNA affinity purification proves the SRSF1 binding to 3’ss.

The authors performed as well U2AF35 knockdown experiments revealing that U2AF35 also binds to 3’ss.

The results are not conclusive because the au. don’t clearly demonstrated if SRSF1 binds to the SRSF1 binding site close to ag/N or it starts a few nucleotides downstream. 

Moreover the au demonstrated that U2AF35 is also involved in the process. However simultaneous binding of SRSF1 and U2AF35 to this sequence is highly improbable

In conclusion the authors suggest that the 5' end of the selected model exons includes enhancer sequence and SRSF1 is one of its recognition factors. 

This study even if inconclusive is relevant because might contribute to the difficult issue of splice site mutation effect prediction.

The paper is clearly written and it reports sufficient information to understand the significance of this basic science results and their translational application.

The only suggestion is to provide more information in the introduction as well as in the discussion regarding the ubiquitous splicing regulator SRSF1.

Reviewer 2 Report

This study aims to shown 5' end of their experimental model exons includes enhancer sequence and SRSF1 is one of its recognition factors. The experimental design is very straightforward. My major concern is only ONE cell line Hela was used in this study. How representative of their results in mammalian cells in general. Minor concern is no Western blotting data to show SRSF1 and U2AF35 were overexpressed for the experiment mentioned at Figure 4.  
